# Mucoadhesive Pharmacology: Latest Clinical Technology in Antiseptic Gels

**DOI:** 10.3390/gels10010023

**Published:** 2023-12-26

**Authors:** María Baus-Domínguez, Felipe-Rodrigo Aguilera, Fernando Vivancos-Cuadras, Lourdes Ferra-Domingo, Daniel Torres-Lagares, José-Luis Gutiérrez-Pérez, Tanya Pereira-Riveros, Teresa Vinuesa, María-Ángeles Serrera-Figallo

**Affiliations:** 1Departamento de Estomatología, Facultad de Odontología, Universidad de Sevilla, 41009 Sevilla, Spain; jlgp@us.es (J.-L.G.-P.); maserrera@us.es (M.-Á.S.-F.); 2Unit of Microbiology, Department of Pathology and Experimental Therapeutics, Faculty of Medicine and Health Sciences, University of Barcelona, L’Hospitalet de Llobregat, 08907 Barcelona, Spain; felipe.aguilera@uach.cl (F.-R.A.); tanyapereirariveros@gmail.com (T.P.-R.); tvinuesa@ub.edu (T.V.); 3School of Dentistry, Faculty of Medicine, Universidad Austral de Chile, Valdivia 5090000, Chile; 4LACER Medical Department, C/Boters, 5, 08290 Cerdanyola del Vallès, Spain; fernando.vivancos@lacer.es; 5LACER R&D/Microbiology Department, 08290 Cerdanyola del Vallès, Spain; lourdes.ferra@lacer.es; 6Hospital Universitario Virgen del Rociío, Universidad de Sevilla, 41013 Sevilla, Spain

**Keywords:** antimicrobial gels, antibacterial gels, chlorhexidine, cymenol, periodontal application, substantivity

## Abstract

Chlorhexidine (CHX) is one of the most widely used antiseptics in the oral cavity due to its high antimicrobial potential. However, many authors have stated that the effect of CHX in nonsurgical periodontal therapy is hampered by its rapid elimination from the oral environment. The aim of this study was to determine the antibacterial efficacy of a new compound of chlorhexidine 0.20% + cymenol (CYM) 0.10% on a multispecies biofilm. For this, an in vitro study was designed using a multispecies biofilm model of *Streptococcus mutans*, *Fusobacterium nucleatum*, *Prevotella intermedia*, and *Porphyromonas gingivalis*. Quantification of the microbial viability of the biofilm was performed using 5-cyano-2,3-ditolyl tetrazolium-chloride (CTC) to calculate the percentage of survival, and the biofilms were observed using a a confocal laser scanning microscopy (CLSM). It was observed that the bactericidal activity of the CHX + cymenol bioadhesive gel was superior to that of the CHX bioadhesive gel, in addition to higher penetrability into the biofilm. Therefore, there was greater elimination of bacterial biofilm with the new compound of chlorhexidine 0.2% plus cymenol 0.1% in a bioadhesive gel form compared to the formulation with only chlorhexidine 0.2% in a bioadhesive gel form.

## 1. Introduction

Chlorhexidine (CHX) is a broad-spectrum antimicrobial agent capable of inhibiting the growth of Gram-negative and Gram-positive bacteria, yeasts, and other microorganisms [1,2,3,4] for a prolonged period of time in the oral cavity due to its powerful adsorption in the teeth [5] and mucosa [3]. It is one of the most effective local antiseptic agents used in the treatment of various oral pathologies [6] because it can change the structure of the oral microbiome [1,7].

CHX, which is a cationic biguanide molecule, is rapidly attracted by the negative charges that characterize the bacterial cell surface [3,6,8], altering the integrity of the bacteria and allowing CHX to penetrate through the inner cell membrane, thus increasing permeability. This leads to a leakage of components of low molecular weight. At this point, the antibacterial action is bacteriostatic and can be interrupted if CHX is eliminated. A stable or increasing concentration of CHX is necessary for it to cause irreversible cell damage [3,8].

CHX has a high substantivity, a quality that allows the persistence of antimicrobial action after the product’s application time. To this end, CHX adheres to oral surfaces, controlling the growth of a new biofilm through its bacteriostatic action [2] at low concentrations and bactericidal action at higher concentrations [3].

Substantivity is determined by the absorption of the product, the duration of its activity, and how it interacts either cooperatively or in conflict with other products or with the oral environment itself [2]. CHX has been reported to exhibit antibacterial activity of around 5 h [3] and substantivity of up to 12 h [3,8]. However, a clinical trial by Naiktari et al. (2018) [8] observed a substantivity of CHX 0.2% up to the seventh hour. This result was corroborated by other researchers, who reported that eating and drinking can significantly decrease substantivity, with complete recovery of flora between the third and seventh hour after application of CHX 0.20%.

Improving the substantivity of a product can lead to a direct increase in its antimicrobial action/efficacy. However, the persistence of a product in the oral cavity depends not only on the characteristics of the compound but also on the way it is applied and the patient’s compliance with the manufacturer’s instructions, which in turn is influenced by the anatomy of the area, speed of rinsing [6], constant movement of the mucous membranes, etc.

The prolonged use of CHX has also been associated with various adverse effects, such as staining, taste alterations, or even the development of oral ulcers [2], as well as potential resistance of oral bacteria when used as a routine antiseptic [2,6].

For all these reasons, work is being conducted to constantly improve the action of an antiseptic compound on microorganisms with the aim of achieving the greatest antibacterial action in the shortest possible time and minimizing the side effects as much as possible. To this end, many studies have focused on optimizing the formulation of chlorhexidine or developing a compound that surpasses or equals its action.

Cymenol (o-cymen-5-ol; C_10_H_14_O) (CYM) is a substituted phenolic compound derived from isopropylene glycol [7] that has demonstrated antimicrobial activity against bacteria, yeasts, and fungi and is safe at concentrations up to 0.1% in cosmetics [4]. To date, it has been used in oral care products as an alternative to chlorhexidine, among other antiseptic compounds [7].

The aim of this study was to experimentally validate the bactericidal activity of a new bioadhesive gel from LACER Laboratories (LACER S.A.U, Barcelona, Spain) consisting of chlorhexidine 0.20% and cymenol 0.10% against the standard chlorhexidine 0.20% gel.

## 2. Results and Discussion

### 2.1. Results on Biofilm Viability Using CTC

The results obtained showed a higher bactericidal activity of the HB31 bioadhesive gel than the CLB bioadhesive gel under the study conditions of 5 min treatment and a concentration of 25% (¼ dilution) of the original product.

The differences in bactericidal activity were calculated on the basis of survival. This showed that the HB31 gel had a much higher activity of 35% mortality compared to the CLB gel, which had only 3% mortality (Table 1), under the aforementioned study conditions.

Nowadays, different compounds with antimycobacterial properties can be found with the aim of reducing the bacterial load or keeping the bacterial biofilm in the oral cavity within the limits of healthiness. Among them, chitosan, for example, has been used in the preparation of oral films [9,10]. Chitosan is a natural polymer that comes from chitin; its qualities include not only its intrinsic bacterial activity against *P. gingivalis* and *Aggregatibacter actinomycetemcomitans* but also its biocompatibility, hemostasis, antioxidant capacity, and mucoadhesiveness [9,10,11,12]. Some studies have stated that chitosan appears to be useful in reducing pain and inflammation, promoting tissue regeneration in the area [11]. Recently, Sáez-Alcaide et al. (2020) [11] marketed a gel containing chlorhexidine 0.2%, chitosan, allantoin, and dexpanthenol and tested it in a clinical trial for the treatment of pain and inflammation after third molar extraction surgery. The authors affirmed that this gel led to a significant reduction in postoperative pain, trismus, and inflammation. However, its comparison was against a placebo and using a subjective scale; in addition, other authors have reported that chitosan does not improve pain in the area [11,12], so the results observed as acceptable or good might not necessarily have come from the action of chitosan or chlorhexidine. Despite the fact that chlorhexidine prevents the accumulation of dental plaque and therefore the oral diseases derived from it, due to its broad spectrum of antibacterial action, minimal systemic absorption, and high substance, its use for prophylactic or therapeutic treatment of mucositis induced by oncotherapy does not present benefits in terms of pain management, although a reduction in oral damage and a lower incidence of bacterial and fungal infections have been observed [13]. These results agree with those previously published by Madrazo-Jiménez et al. (2016) [14] in a prospective open-mouth study, where the authors reported a significant improvement in surgical wound healing but without improvements in swelling and pain. This study also made comparisons with a placebo control group. In line with pain management, it should be noted that there appear to be no significant differences in pain intensity or quality of life when comparing CHX 0.12% rinses versus 0.20% [14].

Following the recommendations of the US Food and Drug Administration and the World Health Organization (WHO), which advise limiting chlorhexidine use to 6 months [15,16] due to the adverse effects associated with its prolonged and constant application, the alternative is to use Salvadora persica, a medicinal plant whose effectiveness is related to the presence of benzyl isothiocyanate. This plant is capable of inhibiting acid production and *S. mutans* growth and has shown antiviral activity against herpes simplex virus (HSV) and antifungal activity against *Candida albicans* [16]. Its antiplaque efficacy against chlorhexidine has been debated, and although there are investigations that report similar results, there are others that support the use of chlorhexidine. Thus, both the systematic review and meta-analysis by Jassoma et al. (2019) [15], like that of Adam et al. (2023) [16], showed evident statistically significant antiplaque and anticariogenic effects compared to the use of a placebo, although with lower efficacy than that observed with chlorhexidine, either at 0.12% or 0.2%.

In the continuous attempt to find a substitute for chlorhexidine and after relying on authors who refute any statistically significant differences in the incidence of alveolar osteitis when using saline rinses vs. chlorhexidine 0.12%, Coello-Gómez et al. carried out a randomized controlled clinical trial in 2018 [17], where the experimental group was treated with Dermacyn Wound Care, a solution of 99.98% superoxidized water and <0.02% of different reactive chlorine and oxygen species. The study confirmed that at least the same results were observed in terms of infectious complications, swelling, pain, and healing, compared to those seen in the CHX group. On the other hand, data have also been presented on the use of particulate hydroxyapatite in the management of oral biofilms with promising results in terms of reduction in bacterial adhesion on the tooth surface, with the effect being comparable to that observed with chlorhexidine in situ but without any antibacterial effect [18].

In this study, a new compound of chlorhexidine + cymenol was used as o-cymen-5-ol had been used to date in combination with zinc chloride (ZnCl_2_) [7,19] or with CPC (cetylpyridinium chloride) due to its potent action against gingival bleeding and plaque accumulation, among others [7]. For its part, CPC has also been tested in different formulations together with different concentrations of CHX. For example, in the in vitro study by Zayed et al. (2022) [20], among other formulations, the authors compared CHX 0.12% + CPC 0.05% rinses of two different brands against CHX 0.20% + ADS (antidiscoloration system) and CHX 0.12% + ADS. It was observed that despite the different concentrations and combinations, there were similar impacts on bacterial survival and the biofilm ecosystem. However, another product with CHX 0.12% + CPC showed a more pronounced effect compared to the previous group. This suggests that although CPC appears to increase the effectiveness of CHX, this is not always the case as its antibacterial activity appears to depend largely on the specific formulation of the product.

O-cymen-5-ol has a direct inhibitory effect against *S. mutans, Actynomices viscosus*, *P. gingivalis*, and *F. nucleatum* [7], which is corroborated by our results as the multispecies biofilm in this study was composed of *S. mutans*, *P. gingivalis*, *P. intermedia*, and *F. nucleatum*. This compound, together with zinc salts, can also inhibit or slow down the growth of other oral pathobionts, such as *Prevotella*, *Actinomyces*, and *Eubacterium*, among others. Similarly, and somewhat innovatively, o-cymen-5-ol does not cause dysbiosis [7]. Furthermore, when the CHX 0.20% + cymenol 0.1% gel was tested on a *S. mutans* biofilm, a total reduction of bacteria was obtained compared to the reduction achieved by the CHX 0.20% gel (Table 2) at 5 min of treatment. Therefore, the CHX + cymenol gel is considered active against *S. mutans* at 1 and 5 min of exposure, compared to CHX alone, which is only considered active when the treatment lasts 5 min. This experiment was conducted prior to the one with the multispecies biofilm because, within the possible models as a basis for testing, the biofilm model generated by *S. mutans* is accepted by the scientific community. At the oral level, *S. mutans* is considered the most cariogenic microorganism in the dental biofilm due to its ability to use carbohydrates such as sucrose to synthesize extracellular polysaccharides (EPSs) and its acidogenic and aciduric properties. EPSs are essential in the virulence of this bacteria as they promote bacterial adherence to the tooth surface and contribute to the structural integrity of dental biofilms [21]. The use of *S. mutans* as an action model is and has been supported as an ideal model to test oral biofilms and the behavior of various substances on them [22,23,24].

### 2.2. CLSM Results Reported

The CLSM images observed confirmed the previously reported results, where a higher bactericidal activity of the HB31 gel versus the CLB gel was observed in the multispecies biofilm studied (*S. mutans*, *P. gingivalis*, *P. intermedia*, and *F. nucleatum*) (Figure 1 and Figure 2).

When evaluating z-stacks using CLSM, the HB31 gel showed a higher penetrability in the multispecies biofilm studied under the treatment conditions established in this study (Figure 3). As shown in the 3D representations of the confocal microscopy, the biofilm treated with the CHX product showed lower antibiofilm effectiveness as its penetration into it was lower, something that was confirmed by the green color still being observed, corresponding to the bacteria that make up the biofilm.

The increased mortality of biofilms treated with HB31 could be due to increased penetration of the gel through the biofilm. This result holds great promise for a chemical agent to aid the treatment of periodontal diseases as traditional therapies have been found to lack the necessary long-term efficacy [6]. Therefore, the current trend is towards the development of a polymer–drug system that allows prolonged release of the active ingredient in a sustained or controlled manner after local application [9]. The main problem encountered in the oral cavity is the early and involuntary elimination of the applied drug due to the peculiarity of the movement and the oral environment itself [6,9]. Therefore, a greater distribution of the compound through the bacterial biofilm would be related to a rapid pharmacological action, reducing the number of microorganisms despite an early elimination of the gel.

Taking into account the available literature, there have been many innovations seeking a greater drug action related to an improvement in the substantivity time. In the case of the study by Al-Ani, E., et al. (2021) [10], mucoadhesive tablets were designed with the premise of a short retention period of chlorhexidine at the oral level in any form of presentation (rinse, spray, or gel). On the other hand, the systematic review and meta-analysis by Zhao, H., Hu, J., and Zhao, L., (2020) [6] stated that no additional benefits were observed in the adjunctive use of CHX after scaling and root planing at the subgingival level or even from the modification of chlorhexidine included in the xanthan gum. Other researchers, such as ZJ. Chen et al. [25], designed a polycaprolactone electrospun nanofiber membrane with sustained chlorhexidine with the aim of achieving sustained drug release. Tarawneh et al. (2021) [26] designed hydrogel films based on sodium carboxymethylcellulose for the administration of CHX in the dentinal tubules with the aim of treating periodontal disease and observed a slow release, indicating the system as a promising option.

For its part, chitosan has also been widely used in the health field as a biomaterial in the design of nanocarriers that improve the bioavailability of a drug in question, as was the case of using psoralidin for the treatment of lung cancer cells [27] or, more specifically at the oral level, for the transport of silver nanoparticles and ibuprofen for the treatment of periodontal disease [9] or chlorhexidine with the aim of improving its antimicrobial effect against planktonic cells and in biofilms [28]. The latest in vitro study by Araujo et al. (2022) [28] indicated that the IONPs-CS-CHX nanocarrier demonstrated more pronounced effects in the reduction of total biomass and biofilm metabolism compared to the exclusive use of CHX, refuting the theory that CHX experiences a reduction in its effectiveness against microorganisms in biofilms because the extracellular matrix barrier impairs the penetration of the drug into the deeper layers.

In this study, the gel form of presentation was not modified as this form of application appears to offer a substantial improvement in bacteria-fighting ability, resulting in a considerable decrease in plaque scores and the gingival index. In some studies, this performance was comparable to that observed with the use of mouthwash [29]. Furthermore, thanks to the combination of chlorhexidine and cymenol, a higher penetration capacity into biofilms was achieved, much more than compared to triclosan or cetylpyridinium chloride [2,7]. The combination of o-cymen-5-ol and CPC also increased the substantivity of CPC by more than one hour as well as reducing bacterial recovery compared to the use of CPC alone.

### 2.3. Limitations

It is crucial to bear in mind that the conditions evaluated in laboratory studies do not accurately reflect real-life situations. Therefore, caution should be exercised when applying the results of these in vitro studies to the reality present in living beings. This is because these studies do not consider factors such as the manner of application by the patient, the effects of saliva rinsing, the influence of various salivary compounds, and the impact of diet, among other relevant aspects.

The in vitro study by Hägi et al. in 2015 [30] provides very relevant conclusions about the nonsurgical treatment of periodontitis as it affirms that a biofilm model that simulates a periodontal pocket allows different treatment modalities to be correctly evaluated; in this case, air polishing with erythritol and CHX achieved greater bacterial reduction. Although the treatment aspect is important, so is the prevention over time as the biofilm is the main etiopathogenetic factor of periodontal pathology.

Consequently, while in vitro studies are valuable for identifying potential compounds of interest, it is imperative to conduct comprehensive clinical trials in vivo to establish effective medical or social protocols.

## 3. Conclusions

The main findings of the paper can be summarized as follows.

The bacterial biofilm penetration of the new compound of chlorhexidine 0.2% plus cymenol 0.1% in a bioadhesive gel form was greater compared to the formulation with only chlorhexidine 0.2% in a bioadhesive gel form.

The antimicrobial activity of the new compound of chlorhexidine 0.2% plus cymenol 0.1% in a bioadhesive gel form was greater compared to the formulation with only chlorhexidine 0.2% in a bioadhesive gel form.

The bacterial biofilm elimination of the new compound of chlorhexidine 0.2% plus cymenol 0.1% in a bioadhesive gel form was greater compared to the formulation with only chlorhexidine 0.2% in a bioadhesive gel form.

The improvement represented by the chlorhexidine + cymenol gel could represent an effectivity increase in terms of treatment of the bacterial biofilm that causes periodontal diseases, making it an adjuvant to treatment. A clinical study is proposed in patients to verify this effectiveness in vivo.

## 4. Materials and Methods

### 4.1. Type of Study

This was an in vitro study aimed at finding out the antimicrobial activity of a new compound (gel) of chlorhexidine 0.20% + cymenol 0.10% (HB31) compared to chlorhexidine 0.20% gel (CLB).

### 4.2. Bacterial Strains and Culture Conditions

The microorganisms selected for this study were representative of those species involved in the etiology of gingivitis.

Standard reference strains of *Streptococcus mutans* NCTC 10449, *Fusobacterium nucleatum* NCTC 10562, *Prevotella intermedia* DSMZ 20706, and *Porphyromonas gingivalis* NCTC 11834 were used. *S. mutans* was grown on Columbia blood agar plates under CO_2_ conditions at 37 °C for 24–48 h, and the rest of the strains were grown on fastidious anaerobe agar plates under anaerobic conditions (10% H_2_, 10% CO_2_, and balance N_2_) at 37 °C for 24–72 h.

### 4.3. Biofilm Development

Biofilm development was performed in sterile polystyrene 24-well tissue culture plates (Greiner Bio-one, Frickenhausen, Germany).

In order to determine the effect of the oral products to be studied on bacterial viability, a multispecies biofilm formation was designed, where the substrates were incubated with poly-L-lysine solution 0.01% (Sigma-Aldrich, Darmstadt, Germany) for 1 h at 37 °C to promote and stabilize the biofilm.

The growth kinetics were evaluated by generating growth curves. The bacterial concentration was adjusted by spectrophotometry (optical density (OD) at 600 nm) in order to obtain a bacterial suspension containing 10^5^ CFU/mL (colony-forming units/mL) for *S. mutans* and 10^6^ CFU/mL each for *F. nucleatum*, *P. gingivalis*, and *P. intermedia*.

The culture medium used for the growth of *S. mutans* was brain–heart infusion broth (Scharlab, Barcelona, Spain) under CO_2_ conditions at 37 °C for 24–48 h.

*F. nucleatum*, *P. intermedia*, and *P. gingivalis* were grown on fastidious anaerobe broth (EO labs, Bonnybridge, UK) under anaerobic conditions at 37 °C for 48–72 h for *F. nucleatum* and 72–96 h for *P. intermedia* and *P. gingivalis*.

### 4.4. Composition of the Gels Studied

The composition of the gel studied of chlorhexidine 0.20% gel (CLB) is detailed in Table 3.

The composition of the gel studied of chlorhexidine 0.20% + cymenol 0.10% (HB31) is detailed in Table 4.

### 4.5. Viability of Biofilm Using 5-5-Cyano-2,3-Ditolyl Tetrazolium-Chloride (CTC)

Once the biofilm was formed, the supernatant was removed from the plates and then washed with ¼ Ringer’s solution to remove any planktonic forms of the study bacteria.

The treatment wells were filled with the oral products to be compared and then incubated for 5 min with a 25% treatment concentration (¼ dilution) of the original product.

After incubation, the product was removed, and the wells were washed again with ¼ Ringer’s solution to remove any remaining product residue.

Finally, the microbial viability of the biofilm was quantified with CTC. For this, 500μL/well of 5 mM CTC was added and incubated overnight to ensure penetration of the CTC through the biofilm matrix. Subsequently, excess CTC was removed again with ¼ Ringer’s solution. The wells were then filled with 1 mL of 96% ethanol to dilute the CTC formazan crystals formed inside the bacteria and left to incubate in the cabinet for 2–3 h, always protected from light.

Finally, quantification was carried out by absorbance at 450 nm, followed by a quick shaking for 30 s and addition of 100 μL/well in a 96-well plate.

This form of quantification is based on the measurement of the metabolic activity of the bacteria. CTC is a water-soluble redox indicator that has the property of changing color depending on the oxidation state.

CTC can be used as an indicator of the microbial metabolic activity of aerobic and strict anaerobic bacteria as they actively reduce CTC during all stages of growth. The reduction of CTC results in CTC formazan, a reddish, highly fluorescent, water-insoluble product that accumulates intracellularly. Its detection, either by measurement of absorbance (at λ = 450 nm) or fluorescence (excitation peak: λ = 480 nm; emission peak: λ = 630 nm), allows the viability of the microorganisms involved in the biofilm after antimicrobial treatment to be determined.

Following treatment of the biofilm, absorbance measurements were taken, and the survival percentages of the treated biofilms were calculated using the following formula:Survival percentage (%) = (Tr − B)/(C − B) ∗ 100(1)

Survival percentage (%) = (Tr − B)/(C − B) ∗ 100

C: absorbance of the control biofilms (mean)

B: absorbance in the absence of biofilm (blank) (mean)

Tr: absorbance of the treated biofilms (mean)

### 4.6. Confocal Laser Scanning Microscopy (CLSM)

This technique makes it possible to detect the light emitted by fluorescent molecules located in the same focal plane and reconstruct three-dimensional images of biological structures.

The idea was to visualize live and dead bacteria present in the biofilms before and after treatment with the oral products under study after staining with Syto 9 and propidium iodide (PI) fluorochromes.

Syto 9 stains both live and dead bacteria, but PI only penetrates microorganisms with membrane alterations and therefore with cell damage, causing a reduction in Syto 9 fluorescence when both dyes are present. Thus, the total number of living microorganisms was determined by the fluorescence emitted by the Syto 9 fluorochrome, and the number of dead microorganisms was obtained by measuring the PI fluorescence.

The observation was performed with a Carl Zeiss CLSM (Carl Zeiss Microscopy GmbH, Jena, Germany) equipped with a 488 nm argon laser and a 543 and 633 nm He/Ne laser using a 63× oil immersion objective (1.4 numerical aperture) with an image resolution of 1024 × 1024 pixels.

In order to carry out this experimental phase, it was necessary to slightly modify the protocol described above. This time, the biofilm was formed on a slide with 8 independent IBIDI wells (μ-Slide 8 Well Glass Bottom, sterilized; Cat. No.: 80827) with 400 μL/well of the starting microbial suspensions. These slides were treated with poly-L-lysine solution to facilitate bacterial adhesion.

After removing the oral products to be studied by rinsing with ¼ Ringer’s solution, staining was carried out. The biofilm was then incubated with a live/dead staining solution (Viability Kit, L7012, Molecular Probes, Invitrogen) prepared according to the manufacturer’s instructions for 30 min in the anaerobic cabinet.

Finally, they were observed under a CLSM. Micrographs were extracted from different fields of the biofilm and from all focal planes of each of the selected fields. Image processing was performed with the ZEN v2.3 software.

The experiment was performed in triplicate.

### 4.7. Statistical Analysis

The data obtained from the bacterial counts were processed using the SPSS software. After performing a descriptive analysis and confirming the normality of the data, the Student’s *t*-test was applied to compare the means of the two study groups (one-tailed distribution, unequal standard deviation, *p*-value < 0.05).

## Figures and Tables

**Figure 1 gels-10-00023-f001:**
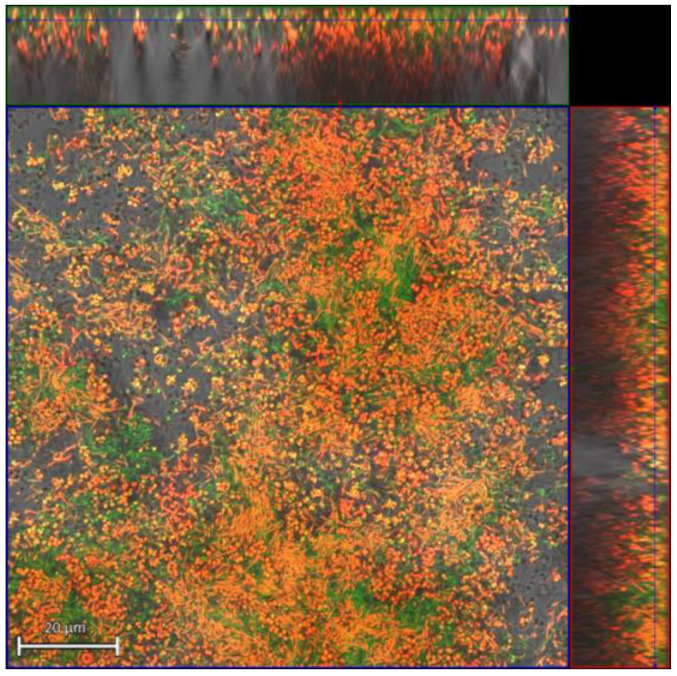
Representative micrographs of bacterial viability using CLSM. Biofilm with application of chlorhexidine 0.20% + cymenol 0.10% gel (HB31). Note: green: viable bacteria; red: dead bacteria; yellow/orange: intermediate colors that can sometimes be observed in bacterial cells stained with a live/dead kit, resulting from the superposition of a green pixel and red pixel in the same z-plane. The yellow cells were considered viable, while orange cells were considered damaged. Scale bar = 20 μm.

**Figure 2 gels-10-00023-f002:**
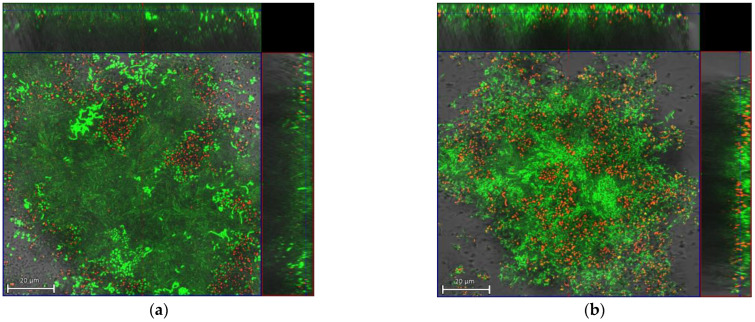
Representative micrographs of bacterial viability using CLSM: (**a**) biofilm control; (**b**) biofilm with application of chlorhexidine 0.20% gel (CLB). Note: green: viable bacteria; red: dead bacteria; yellow/orange: intermediate colors. Scale bar = 20 μm.

**Figure 3 gels-10-00023-f003:**
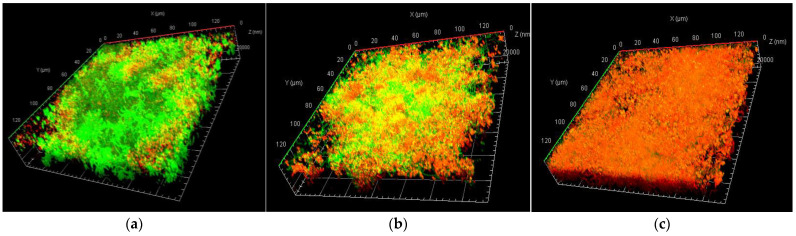
Representative 3D micrographs of bacterial viability using CLSM: (**a**) biofilm control; (**b**) chlorhexidine 0.20% gel (CLB); (**c**) chlorhexidine 0.20% + cymenol 0.10% gel (HB31). Note: green: viable bacteria; red: dead bacteria; yellow/orange: intermediate colors.

**Table 1 gels-10-00023-t001:** Results of the bacterial survival of the biofilm.

Compound	% Bacterial Viability	Standard Deviation
HB31	64.94	0.03
CLB	97.10	0.33

HB31: chlorhexidine 0.20% + cymenol 0.10%; CLB: chlorhexidine 0.20%. The data obtained from the bacterial counts were processed using SPSS software v.27. After performing a descriptive analysis and confirming the normality of the data, the Student’s *t*-test was applied to compare the means of the two study groups.

**Table 2 gels-10-00023-t002:** Comparative bactericidal activity of CHX 0.20% vs. CHX 0.20% + CYM 0.10%.

Product	Contact Time	Initial Count Number (cfu/mL)	Initial Count (log_10_ cfu/mL)	Count Number after t Exposure (cfu/mL)	Count after t Exposure (log_10_ cfu/mL)	R = Bactericidal Activity (log_10_)
Chlorhexidine 0.20%	1 min	1.5 × 10^7^	7.18	5.0 × 10^2^	2.70	4.48
	5 min			7.0 × 10^1^	1.90	5.28
Chlorhexidine 0.20% + CYM 0.10%	1 min	1.5 × 10^7^	7.18	4.1 × 10^1^	1.61	5.56
	5 min			<10	0.00	6.18

<10 CFU/mL: detection limit of the technique.

**Table 3 gels-10-00023-t003:** Chlorhexidine bioadhesive gel composition (CLB).

LACER Name	% (m/m)
Toothpaste aroma 1/074569	0.12
Chlorhexidine digluconate 20%	1.06
Potassium acesulfame	0.08
Purified water	71.315
Methyl salicylate	0.025
Propilenglicol	10.00
Menthol crystal	0.10
Natrosol Pharma (hydroxyethyl cellulose)	3.00
Glycerin	7.00
Sorbitol (bulk)	7.00
Peg40 hydro castor oil (cosmetics)	0.30

pH = 5.90–7.30; density 20 °C: 1.048–1.068 g/mL; record type: dent.

**Table 4 gels-10-00023-t004:** Chlorhexidine + cymenol bioadhesive gel composition (HB31).

LACER Name	% (m/m)
Toothpaste aroma 1/074569	0.12
Chlorhexidine digluconate 20%	1.06
O-cymen-5-ol	0.10
Purified water	71.047
Sucralose	0.07
Propilenglicol	10.00
Vanilla	0.003
Menthol crystal	0.10
Benecel K4M (hydroxypropyl methyl cellulose)	3.20
Glycerin	7.00
Sorbitol (bulk)	7.00
Peg40 hydro castor oil (cosmetics)	0.30

pH = 5.9–7.2; density 20 °C: 1.052 g/mL ± 0.010 g/mL (1.042–1.062 g/mL).

## Data Availability

The data presented in this study are available on request from the corresponding author. The data are not publicly available due to patent restrictions.

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
