# Peer review of "Mucoadhesive Pharmacology: Latest Clinical Technology in Antiseptic Gels"

_gels, 2023, doi:10.3390/gels10010023_

Round 1
Reviewer 1 Report
Comments and Suggestions for Authors
The manuscript must be improved, as follows:
- Acronym must be explained when firstly introduced (e.g. Chlorhexidine (CHX), etc)
- The terms “(1) Background:”, (2) Methods:, (3) Results: and (4) Conclusions:” should be removed from the abstract. Also the sentences must have a sense in standalone form, for example “Bactericidal activity of the CHX + CYM bioadhesive gel was found to be superior to that of the CHX bioadhesive gel, in addition to higher penetrability into the biofilm”.
- Please explain clearly what is “LACER”? It is a commercial name of a product (0.20% Chlorhexidine and 0.10% Cymenol) ? if yes, this must be accompanied by additional details (as for any other chemicals).
- Why the standard deviation (Table 1) is one order of magnitude higher for CHX in comparison with CHX + CYM?
- The sentence “Among them, chitosan (CS), for example, has been used in the preparation of oral films [9,10]”, was written two times in the text.
- In pag 3, row 97, “Actinomycetecomitans” is the same as “Aggregatibacter actinomycetemcomitans”?
- The results presented in Table 2 must be detailed (experimental details about S. mutans biofilm). Also explain why the bactericidal activity was tested only on “S. mutans”?
- In Table 2, “bactericial activity” should be “bactericidal activity”, also “UFC” must be written as CFU (colony-forming unit)
- What is the detailed composition of “multispecies biofilm studied” (pag 4, row 172)
- If there are any other experimental observations (optical density measurements, etc) besides the numerical data presented in Tables 1 and 2, the authors are kindly asked to include them in the manuscript.
- The scale-bar of Fig 1 and Fig 2 is missing.
- The interpretation of Figs 1-3 is missing. Please explain to the readership what the color gradient meaning is and how the images of the Confocal microscopy can be understood? Different colors are present but not explained.
- The authors stated that “The HB31 gel shows a higher penetrability in the multispecies biofilm studied under the treatment conditions established in this study (Figure 3).” Why? Explanations based on Fig 3 are needed.
- The formulation (preparation) of the gels based on Chlorhexidine and Cymenol must be provided (in fact this is reason for paper submission to “Gels” Journal). Some properties of the obtained gels must also be presented. More than this, the authors stated that the gels are “bioadhesive”. Why? There is any experimental procedure related to the adhesion of the gels to the oral tissue?
- The “Statistical Analysis” is not evidenced in the manuscript (just mentioned in the Experimental section)
- The Conclusion Section must be improved (highlighting the main benefits of the research conducted).
Comments on the Quality of English LanguageEnglish is fine
Author Response
Dear Editor and Reviewers,
We thank the time spent reviewing our research article entitled “Mucoadhesive Pharmacology: Latest Clinical Technology in Antiseptic Gels”. Thank you for your comments on our study. We truly believe that it is a very interesting and novel line of research, since there is no published research on the combination of chlorhexidine with cymenol.
We hope that we have answered your questions correctly so that the validity of our work is not called into question. We have been correcting our manuscript point by point, following your recommendations.
We will now proceed to respond to your reviews:
Acronym must be explained whe firstly introduced (e.g. Chlorhexidine (CHX), etc).
Acronyms have been introduced next to the written word for the first time. Thanks for your consideration.
The terms “(1) Background:, (2) Methods:, (3) Results: and (4) Conclusions:” should be removed from the abstract. Also the sentences must have a sense in standalone form, for example “Bactericidal activity of the CHX + CYM bioadhesive gel was found to be superior to that of the CHX bioadhesive gel, in addition to higher penetrability into the biofilm”.
We have removed the headings from each section of the abstract and rewritten it so that each sentence makes sense on its own. Thank you very much for your recommendation.
Please explain clearly whay is “LACER”? It is a commercial name of a product (0.20% Chlorhexidine and 0.10% Cymenol)? If yes, this must be accompanied by additional details (as for any other chemicals).
LACER, S.A.U. is a Spanish pharmaceutical laboratory, and it has been better specified at the end of the introduction of the manuscript. Thank you very much for your appreciation.
Why the standard deviation (Table 1) is one order of magnitude higher for CHX in comparison with CHX + CYM?
Because the variation or dispersion by which individual data points in the CHX sample differ more from the mean than individual data in the CHX + CYM sample.
The sentence “Among the, chitosan (CS), for example, has been used in the preparation of oral films [9,10], was written two lines in the text.
We are very sorry for this error. We have removed the duplicate phrase. Thank you very much for your appreciation.
In pag 3, row 97, “Actinomycetecomitans” is the same as “Aggregatibacter actinomycetecomitans”?
Yes, it is the same. We have added the full name of the bacteria. Thanks for your appreciation.
The results presented in Table 2 must be detailed (experimental details about s. mutans biofilm). Also explain why the bactericidal activity was tested only on S.mutans”?
The experiment on s.mutans was prior to that of the multispecies biofilm, the process was the same as that for the multispecies biofilm. We have added an explanation as to why it has previously been tested on S. mutans. Thanks for your consideration.
In table 2, “bactericidal activity” should be “bactericidal activity” also “UFC” must be written as CFU (colony-forming unit)
We are very sorry for the UFC typo, it has already been corrected to CFU. However, we do not understand the correction about “bactericidal activity”. Thank you very much.
What is the detailed composition of “multispecies biofilm studied” (pag 4, row 172).
We detailed in previous paragraphs the composition of the multispecies biofilm studied; however, we take your assessment into account and we have also added it in this sentence that you indicate. Thank you so much.
If there are my other experimental observations (optical density measurements, etc) besides the numerical data presented in Tables 1 and 2, the authors are kindly asked to include them in the manuscript.
All experimental observations made have been added. Thank you very much for your recommendation.
The scale-bar of Fig 1 and Fig 2 is missing.
Scale has been added to all figures. Thank you so much.
The interpretation of Figs 1-3 is missing. Please explain to the readership what the color gradient meaning is and how the images of the Confocal microscopy can be understood? Different colors are present but not explained.
The interpretation of the color gradient has been explained. Thanks for your consideration.
The authors stated that “The HB31 gel shows a higher penetrability in the multispecies biofilm studied under the treatment conditions established in this study (Figure 3). Why? Explanations based of Fig 3 are needed.
Explanations based on figure 3 have been added. Thank you very much.
The formulation (preparation) of the gels based on Chlorhexidine and Cymenol must be provided (in fact this reason for paper submission to “Gels” Journal). Some properties of the obtained gels must also be presented. More than this, the authors stated that the gels are “bioadhesive”. Why? There is any experimental procedure related to the adhesion of the gels to the oral tissue?
An in vitro mucoadhesivity study was conducted at the Centre for Industrial Rheology to demonstrate that synergistic interactions between polymers and mucin contribute to mucoadhesion. The objective of the study was the following: simple rheological investigation of various dental gels, when combined with mucin solutions, to provide an in vitro prediction of this behaviour. The conclusion was that the Chlorhexidine Gel containing 0.1% Cymenol showed the highest relative interaction with the mucin solution, showing more mucoadhesion.
Tables have been added that refer to the composition of the gels studied.
Both the CHX gel and the CHX + CYM gel are bioadhesive, since CHX presents the property of high substantivity that has been reported in numerous clinical studies in humans, including those referenced in this manuscript. We must not forget that this is an in vitro study, and we declare this in the limitations section of the study. Therefore, the next step will be to test it in a human study.
Thanks for your recommendation.
The statistical Anaysis is not evidenced in the manuscripts (just mentioned in the experimental section).
The data obtained from the bacterial counts were processed using SPSS software. After performing a descriptive analysis and confirming the normality of the data, the Student’s t-test was applied to compare the means of the two study groups.
All variables were normal. If you wish to have the data from the normality study of each variable, we can attach it as an annex, although we do not usually put it in our main manuscript.
We have added the information to the main text under table 1.
The conclusion section must be improved (highlighting the main benefits of the research conducted).
We have added a paragraph to highlight the clinical improvement that the use of the gel would bring. Thanks for your recommendation.
We hope we have responded correctly to this review.
Thank you for your consideration,
Sincerely.
Reviewer 2 Report
Comments and Suggestions for Authors
The manuscript entitled " Mucoadhesive Pharmacology: Latest Clinical Technology in Antiseptic Gels" tries to evaluate the the antibacterial efficacy of a new compound of chlorhexidine0.20% + cymenol 0.10% on a multispecies biofilm.
General remarks:
It is important to add in the manuscript the formation of films and their physico-chemical and pharmacotechnical evaluation.
The manuscript seems to be incomplete.
This phrase appears for two times :"Among them, chitosan (CS), for example, has been used in the preparation of oral films [9,10]."
Author Response
Dear Editor and Reviewers,
We thank the time spent reviewing our research article entitled “Mucoadhesive Pharmacology: Latest Clinical Technology in Antiseptic Gels”. Thank you for your comments on our study. We truly believe that it is a very interesting and novel line of research, since there is no published research on the combination of chlorhexidine with cymenol.
We hope that we have answered your questions correctly so that the validity of our work is not called into question. We have been correcting our manuscript point by point, following your recommendations.
We will now proceed to respond to your reviews:
The manuscript entitled “Mucoadhesive Pharmacology: Latest Clinical Technology in Antiseptic Gels” tris to evaluate the antibacterial efficacy of a new compound of chlorhexidine 0.20% + cymenol 0.10% on a multispecies biofilm.
It is important to add in the manuscript the formation of films and their physico-chemical and pharmacotechnical evaluation. The manuscript seems to be incomplete.
An in vitro mucoadhesivity study was conducted at the Centre for Industrial Rheology to demonstrate that synergistic interactions between polymers and mucin contribute to mucoadhesion. The objective of the study was the following: simple rheological investigation of various dental gels, when combined with mucin solutions, to provide an in vitro prediction of this behaviour. The conclusion was that the Chlorhexidine Gel containing 0.1% Cymenol showed the highest relative interaction with the mucin solution, showing more mucoadhesion.
Tables have been added that refer to the composition of the gels studied.
Thanks for your recommendation.
This phares appears for two times: “Among them, chitosan (CS) for example, has been used in the preparation of oral films”.
We are very sorry for this error. We have removed the duplicate phrase. Thank you very much for your appreciation.
We hope we have responded correctly to this review.
Thank you for your consideration,
Sincerely.
Reviewer 3 Report
Comments and Suggestions for Authors
The topic raised in the article is interesting from the point of view of practical dentistry. Although, many works are devoted to the search for an effective drug to prevent the growth of biofilms.
The article has a clear structure. However, there are a number of shortcomings that need to be addressed.
1. In the captions to the figures, you need to indicate what staining was used, what exactly we observe to make the material easier to perceive. In addition, it would be advisable to combine the pictures and give photographs separately by channels in addition to the merged images.
2. It is completely unclear what kind of bioadhesive gel was used, what is its basis? Is this a commercial gel or a gel created by the authors?
3. In addition to the method based on biofilm visualization, I would recommend to authors additionally use a culture method to determine changes in the number of colony forming units in the biofilm
4. Specify, please, the name of the confocal microscope and the manufacturer.
Author Response
Dear Editor and Reviewers,
We thank the time spent reviewing our research article entitled “Mucoadhesive Pharmacology: Latest Clinical Technology in Antiseptic Gels”. Thank you for your comments on our study. We truly believe that it is a very interesting and novel line of research, since there is no published research on the combination of chlorhexidine with cymenol.
We hope that we have answered your questions correctly so that the validity of our work is not called into question. We have been correcting our manuscript point by point, following your recommendations.
We will now proceed to respond to your reviews:
In the captions to the figures, you need to indicate what staining was used, what exactly we observe to make the material easier to perceive. In addition, it would be advisable to be combine the pictures and give photographs separately by channels in addition to the merged images.
The scales and interpretation of the color gradient have been added to the legends of the figures for better understanding by readers. Thanks for your consideration.
It is completely unclear what kind of bioadhesive gel was used, what is its basis? Is this a commercial gel or a gel created by the authors?
At the time of the study the 0.2% chlorhexidine gel was a commercial gel and the gel incorporating 0.1% Cymenol was an experimental prototype. Thanks you.
In addition to the method based on biofilm visualization, I would recommend to the authors additionally use a culture method to determine changes in the number of colony forming units in the biofilm.
Only one culture method was used to determine changes in the number of colony-forming units in the biofilm of S. mutans. This additional method was not used for the multi-species biofilm study, but we will take it into account for future studies. Thank you very much for your recommendation.
Specify, please the name of the confocal microscope and the manufacturer.
We have added it in section 4.3. Confocal microscopy "Carl Zeiss CLSM (Carl Zeiss Microscopy GmbH, Jena, Germany)". Thanks for your recommendation.
We hope we have responded correctly to this review.
Thank you for your consideration,
Sincerely.
Round 2
Reviewer 1 Report
Comments and Suggestions for Authors
The paper can be accepted for publication after checking the English in the whole manuscript. Two examples:
(1) In the next sentence, the "in vitro study" is repeated: For this, an in vitro study was designed in vitro study on multispecies biofilm Streptococcus mutans, Fusobacterium nucleatum, Prevotella intermedia and Porphyromonas gingivalis
(2) In the next sentence, the "improvment" is repeated: The improvement represented by the Chlorhexidine + Cymenol gel could represent an improvement in terms of the treatment of the bacterial biofilm that causes periodontal diseases, as an adjuvant to treatment, and maintaining the results. A clinical study is proposed in patients to verify this effectiveness in vivo.
Also, the first three phrases in the Conclusion section must be reformulated to have sense in standalone form or a sentence should be introduced like "The main findings of the paper are:"
Comments on the Quality of English LanguageThe paper can be accepted for publication, after solving the following issues:
The English must be checked in the whole manuscript. Two examples:
(1) In the next sentence, the "in vitro study" is repeated: For this, an in vitro study was designed in vitro study on multispecies biofilm Streptococcus mutans, Fusobacterium nucleatum, Prevotella intermedia and Porphyromonas gingivalis
(2) In the next sentence, the "improvment" is repeated: The improvement represented by the Chlorhexidine + Cymenol gel could represent an improvement in terms of the treatment of the bacterial biofilm that causes periodontal diseases, as an adjuvant to treatment, and maintaining the results. A clinical study is proposed in patients to verify this effectiveness in vivo.
Also, the first three phrases in the Conclusion section must be reformulated to have sense in standalone form or a sentence should be introduced like "The main findings of the paper are:"
Author Response
Dear Editor and Reviewers,
We are very grateful for your comments on our research article “Mucoadhesive Pharmacology: Latest Clinical Technology in Antiseptic Gels”. We continue to improve the manuscript on the basis of your corrections and recommendations.
We will now proceed to respond to your reviews:
The paper can be accepted for publication after checking the English in the whole manuscript. Two examples:
(1) In the next sentence, the "in vitro study" is repeated: For this, an in vitro study was designed in vitro study on multispecies biofilm Streptococcus mutans, Fusobacterium nucleatum, Prevotella intermedia and Porphyromonas gingivalis
(2) In the next sentence, the "improvment" is repeated: The improvement represented by the Chlorhexidine + Cymenol gel could represent an improvement in terms of the treatment of the bacterial biofilm that causes periodontal diseases, as an adjuvant to treatment, and maintaining the results. A clinical study is proposed in patients to verify this effectiveness in vivo.
Also, the first three phrases in the Conclusion section must be reformulated to have sense in standalone form or a sentence should be introduced like "The main findings of the paper are:"
à We have corrected our manuscript according to your recommendations. Thank you so much.
We hope we have responded correctly to this review.
Thank you for your consideration,
Sincerely.
Reviewer 2 Report
Comments and Suggestions for Authors
The authors said that made " simple rheological investigation of various dental gels".
Please add the method and results.
Author Response
Dear Editor and Reviewers,
We are very grateful for your comments on our research article “Mucoadhesive Pharmacology: Latest Clinical Technology in Antiseptic Gels”. We continue to improve the manuscript on the basis of your corrections and recommendations.
We will now proceed to respond to your reviews:
The authors said that made "simple rheological investigation of various dental gels".
Please add the method and results.
à Dear reviewer, the simple rheological investigation of the dental gels used in this study precedes the marketing of the gels. Both gels used in this study are already on the market. The objective of this manuscript is not a study of the rheology of the gels used, but to experimentally (in vitro) validate the bactericidal activity of the new bioadhesive gel of LACER Laboratories (LACER SAU, Barcelona, Spain) of Chlorhexidine 0.20% and Cymenol 0.10% against the standard Chlorhexidine 0.20% gel. We have added all the information on how the cultivation and development of the biofilm was carried out, as well as the composition of the gels studied and the bacterial viability after the application of the gels, in the Materials and Methods section.
In any case, we have planned a study in humans on the improved formula of Chlorhexidine 0.20% and Cymenol 0.10%, with the aim of knowing the substantivity (long-lasting effect), among others, in which we are currently working.
Again, thank you very much for your comments.
We hope we have responded correctly to this review.
Thank you for your consideration,
Sincerely.
Reviewer 3 Report
Comments and Suggestions for Authors
The authors answered all the questions and explained the unclear points
Author Response
Thankyou